# The Association between Utilization of Media Information and Current Health Anxiety Among the Fukushima Daiichi Nuclear Disaster Evacuees

**DOI:** 10.3390/ijerph17113921

**Published:** 2020-06-01

**Authors:** Masatsugu Orui, Chihiro Nakayama, Yujiro Kuroda, Nobuaki Moriyama, Hajime Iwasa, Teruko Horiuchi, Takeo Nakayama, Minoru Sugita, Seiji Yasumura

**Affiliations:** 1Department of Public Health, Fukushima Medical University School of Medicine, Fukushima 960-1295, Japan; nakac@fmu.ac.jp (C.N.); moriyama@fmu.ac.jp (N.M.); hajimei@fmu.ac.jp (H.I.); t-hori@fmu.ac.jp (T.H.); yasumura@fmu.ac.jp (S.Y.); 2Sendai City Mental Health and Welfare Center, Sendai 980-0845, Japan; 3Center for Integrated Science and Humanities, Fukushima Medical University, Fukushima 960-1295, Japan; kuroday@fmu.ac.jp; 4Tokyo Metropolitan Institute of Gerontology, Tokyo 173-0015, Japan; 5Department of Health Informatics, School of Public Health, Kyoto University, Kyoto 606-8501, Japan; nakayama.takeo.4a@kyoto-u.ac.jp; 6Toho University, Tokyo 143-8540, Japan; sugitamnr@a05.itscom.net

**Keywords:** Fukushima nuclear accident, mass media, Internet, public health practice, community mental health services

## Abstract

The 2011 nuclear disaster in Fukushima was not only a health disaster, but also an information disaster. Although media can promote health communication following disasters, studies have revealed associations between media information and negative psychological reactions. To clarify the relationship between media utilization and current health anxiety due to radiation exposure, a cross-sectional questionnaire survey was conducted in Fukushima. We selected 2000 subjects from evacuation (i.e., 500) and non-evacuation (i.e., 1500) areas by two-stage stratified random sampling. As the independent variable, participants were asked about current health anxiety due to radiation exposure at the time of answering the questionnaire. For utilization of media about radiation exposure, local media, national media, Internet media, public broadcasts, and public relations information from local government were set as the dependent variables. Questionnaire data were analyzed by evacuation type (i.e., forced/voluntary). In a multivariate logistic regression analysis, the use of public relations information was significantly associated with lower anxiety for the forced evacuees (odds ratio: 0.72; 95% confidence interval: 0.56–0.93). Our findings highlight the importance of public relations information from local government in terms of it being associated with lower current health anxiety, and this could potentially aid in preparing for future disasters.

## 1. Introduction

The Great East Japan Earthquake, which occurred on 11 March 2011, was the largest earthquake ever recorded in Japan’s history. The earthquake (magnitude 9.0) generated a massive tsunami that caused enormous damage to the Pacific Coast. This was followed by a separate tsunami, which hit the Fukushima Daiichi Nuclear Power Plant operated by the Tokyo Electric Power Company, causing radiation disasters in Fukushima Prefecture and requiring the long-term evacuation of residents from many surrounding municipalities. Due to this triple disaster, more than 92,000 residents who lived in an area designated by the national government as an evacuation area were forced to leave their homes (as of May 2016) [1]. Moreover, some residents decided to evacuate voluntarily to avoid the effects of the nuclear disaster, even residents who lived in non-evacuation areas.

The nuclear accident at the Fukushima Daiichi Nuclear Power Station caused multiple public health problems, including increased anxiety and mental health issues due to perceived risk among the evacuees and residents of Fukushima. In addition, Yamashita, who supported the nuclear accident response on-site as a radiation specialist, have argued that the Fukushima event was not only a health disaster, but also an information disaster [2], because the accident was an unprecedented experience for evacuees, and their perceived radiation exposure risk may have been related to the mass media. Consequently, their disaster-related stress and/or psychological distress levels may have been affected [3]. Indeed, newspaper coverage of the accident focused mainly on the crisis response relating to immediate issues, actions, and decisions in the aftermath of the accident (e.g., information of on-site actions undertaken, communications about the INES (International Nuclear Event Scale), food restrictions, cost, and number of people affected and being evacuated) [4].

One of the recommendations of the Chernobyl Forum report was to address the lack of accurate information available to local populations on the health risks as a result of the disaster itself, as well as wider health risks such as non-communicable diseases [5]. Moreover, the United Nations Sendai Framework for Disaster Risk Reduction aims to understand disaster risk while sharing non-sensitive information and appropriate communications, and to strengthen the utilization of media, including social media and traditional media [5]. In fact, the media functioned as a form of interpersonal communication with others, or as a channel for local government and other organizations during the immediate aftermath of the Great East Japan Earthquake [6]. However, the media are not always helpful. Several studies have examined disaster-related television viewing in the context of terrorism and have explored a range of outcomes, including post-traumatic stress disorder (PTSD), depression, anxiety, stress reactions, and substance use [7]. One study reported a significant association between the consumption of television and Internet coverage of the 2011 Great East Japan Earthquake and Tsunami, and post-traumatic reactions [8]. This suggests that the media (including Internet and television) can trigger negative psychological responses in evacuees and residents who use it.

Against this backdrop, the present study aims to clarify the association between media utilization (e.g., Internet media, public relations information from local government, and other traditional media) and current strong health anxiety at the time of the survey in the context of the Fukushima nuclear disaster, in order to consider effective modes of disaster communication among evacuees. These findings will likely be useful for future disaster risk reduction and management.

## 2. Materials and Methods

### 2.1. Participants

This cross-sectional questionnaire survey targeted 2000 residents of Fukushima Prefecture aged 20–79 years. Participant selection was based on two-stage stratified random sampling (stage one, survey of the region; stage two, survey of individuals). A random selection occurred of 33–34 individuals per point from municipal resident registration files to obtain 2000 representative participants. Of the 2000 subjects, 500 were from the three types of evacuation area that the Japanese government designated according to spatial radiation dose rates, as follows: (1) difficult-to-return areas, with a radiation dose rate ≥50 millisieverts (mSv) per year; (2) residence restriction areas, with a radiation dose rate ≥20 and <50 mSv per year; and (3) areas where evacuation orders were ready to be lifted as of 22 April 2011. The remaining 1500 people lived in the non-evacuation areas of Fukushima Prefecture (500 people were selected from each of the three areas of Hama-Dori, Naka-Dori, and Aizu) (Figure 1). We sent an anonymous, self-reporting postal questionnaire to participants between August and October 2016. The survey was approved by the ethics review committee of Fukushima Medical University on 12 April 2016 (approval number: 2699).

### 2.2. Survey Variables

For the independent variable, i.e., current anxiety regarding perceived radiation health risks, participants were asked to subjectively rate at the time of answering the questionnaire “Your current level of anxiety about the effects of radiation on your health due to the nuclear disaster” on a five-point scale: “Not at all,” “Only a little,” “Somewhat,” “Very,” and “Extremely.” “Very” and “Extremely” were categorized as the “current strong anxiety group,” with the other levels of anxiety as the “no or weak anxiety group.” This questionnaire was investigator-designed.

For utilization of media about radiation, respondents selected up to three items from the following 13 options: local newspapers, national newspapers, NHK (Nippon Hoso Kyokai) television (public broadcast television, both national and local), private local broadcast television, private national broadcast television, radio, Internet news, Internet sites/blogs, social network services (SNS), magazines/books, public relations information from local government, word of mouth, and none of the above. To assess the association between media utilization and current strong health anxiety, we categorized “any local media (local newspapers and broadcasting),” “any national media (national newspapers and broadcasting),” “public broadcasting (NHK),” “any Internet media (Internet news, Internet sites/blogs, SNS),” and “public relations information from local government” as dependent variables, since these types of media were utilized by a relatively large number of respondents.

Regarding current health anxiety due to radiation exposure, participants were asked about: (1) anxiety related to delayed effects (e.g., severe diseases) with the statement “I am worried I might suffer from serious diseases due to the influence of radiation in the future”; (2) anxiety related to unhealthy status with the statement “Every time my condition gets worse, I become anxious about radiation exposure”; (3) anxiety related to genetic effects with the statement “I am worried that the influence of radiation will be inherited by the next generation, such as my children and grandchildren”; and (4) anxiety relating to broadcasting about nuclear issues with the statement “Looking at reports on nuclear power plant accidents, I become very anxious.” These four single-item questions were part of a reliable questionnaire regarding radiation anxiety (i.e., the 7-item Radiation Anxiety Scale developed by Umeda et al. [9] and presented by Fukasawa et al. [10]). The Cronbach’s alpha coefficient of the scale has been reported as 0.81, and in the present study sample, it was 0.84.

The other questionnaire than the 7-item Radiation Anxiety Scale was investigator-designed. All questionnaire items were shown in a previous report presented by Nakayama et al. [11].

### 2.3. Statistical Analysis

Data were also analyzed by evacuation type: (1) forced evacuation (or forced evacuees), which refers to evacuation due to living in an area designated by the national government as an evacuation area, as of 11 March 2011; and (2) voluntary evacuation (or voluntary evacuees), which refers to voluntary evacuation to avoid the effects of the nuclear disaster, even among residents living in non-evacuation areas, as of 11 March 2011. The chi-square test and multivariate logistic regression analysis were used to examine the association between media utilization and current strong health anxiety due to the nuclear disaster, as well as the characteristics of current strong health anxiety among evacuees by evacuation type. Statistical significance was evaluated using two-sided, design-based tests with a 5% level of significance. All statistical analyses were performed using SPSS 23.0 (IBM Corp., Armonk, NY, USA).

## 3. Results

### 3.1. Participants

We sent out 1985 questionnaires (excluding those returned to the sender due to no one residing at the address) and received 916 responses from August to December 2016 (response rate, 46.1%). After excluding 55 respondents who failed to provide information regarding sex or age, as well as 636 respondents who were not evacuees or did not answer a question about relocation due to nuclear disaster, the final study population consisted of 225 respondents who were either forced (*n* = 156) or voluntary (*n* = 69) evacuees (Figure 2).

### 3.2. Respondent Characteristics

The proportions of respondents aged 65 years and older, of respondents with a junior/senior high school education, and of respondents who were unemployed were higher among forced evacuees compared to voluntary evacuees (Table 1).

### 3.3. Utilization of Media Relating to Nuclear Exposure

The type of media with the highest utilization rate was any local media (69.8%), followed by public broadcasting (NHK) (45.3%), and then public relations information from local government (44.0%). There was no significant difference in the utilization of local, national, or public broadcasting (NHK) between forced and voluntary evacuees. In contrast, the utilization rate of Internet media and public relations information from local governments differed significantly between forced and voluntary evacuees (Table 2). Moreover, the characteristics of the users of media relating to nuclear exposure are shown in Appendix A. The Internet media users in this study tended to be of a younger generation and of a higher educational level than users of the other types of media. Furthermore, the proportion of those who utilized any Internet media among voluntary evacuees was higher in comparison to forced evacuees.

### 3.4. Specifics of Current Strong Anxiety

The proportion of respondents with current strong health anxiety due to radiation exposure at the time of answering the questionnaire was 20.3% (43/223). Among evacuees who expressed current health anxiety at the time of answering the questionnaire, most were concerned about the delayed effects (92.9%), the genetic effects (92.9%), and the broadcasting about nuclear issues (95.3%). Only anxiety about unhealthy status was of relatively low concern among evacuees (63.4%). The proportion of evacuees with these concerns was significantly higher among those who expressed current strong health anxiety compared to those who did not. There was no significant difference between forced and voluntary evacuees (Table 3).

### 3.5. Association between Current Strong Health Anxiety and Utilization of Media Information

Among all evacuees, a significant negative association was observed between utilization of public relations information from local government and current strong health anxiety at the time of answering the questionnaire. Among the voluntary evacuees, there was a non-significant trend between utilization of Internet media and current strong health anxiety (Table 4).

In the multivariate logistic regression analysis, utilization of public relations information from local government was significantly associated with lower current strong health anxiety at the time of answering the questionnaire among all evacuees (odds ratio (OR): 0.76; 95% confidence interval (CI): 0.61–0.94) and among forced evacuees (OR: 0.72; 95% CI: 0.56–0.93). However, public broadcasting (NHK) showed a non-significant relation between utilization and lower current health anxiety (OR: 0.85; 95% CI: 0.69–1.04). Moreover, there was a non-significant trend between utilization of Internet media and current strong health anxiety (OR: 1.56; 95% CI: 0.99–2.43) (Table 5).

## 4. Discussion

The present study aimed to clarify the association between media utilization (e.g., Internet media, public relations information from local government, and other traditional media) and current strong health anxiety at the time of answering the questionnaire in the context of the Fukushima nuclear disaster. As per the results, the present study found a significant association between the use of public relations information from local government and lower current health anxiety at the time of answering the questionnaire.

### 4.1. Utilization of Media Information

In a previous study that assessed media consumption after the Fukushima Daiichi nuclear disaster, over 95% of participants answered that they used television news as a common media source, whereas Internet news and personal Internet websites were used by less than 50% (39% and 14%, respectively) [12]. Although a simple comparison with our results is not possible due to differences in the survey items and methods, those findings are largely consistent with our present findings. On the other hand, the rate of use of any Internet media among voluntary evacuees was significantly higher than that among forced evacuees. In fact, the rate of Internet media usage was the highest among the media sources in the voluntary evacuees, while it was the lowest in the forced evacuees. When considering age, the Internet media utilization rate in the voluntary evacuees was 81.3% (13/16 respondents) among those aged 20–39 years (forced: 43.8%; χ^2^ test, *p* = 0.03), 47.1% (16/34 respondents) among those aged 40–64 years (forced: 21.1%; χ^2^ test, *p* = 0.01), and 5.3% (1/19 respondents) among those aged ≥65 years (forced: 1.6%; χ^2^ test, *p* = 0.36) (Table 1 and Appendix A). This suggests that younger generations use Internet media to the greatest extent among age groups, particularly among voluntary evacuees.

The utilization rate of public relations information from local government among forced evacuees was higher than that of voluntary evacuees. Compared by age group, the utilization rate among forced evacuees aged 40–64 years was significantly higher than that of the corresponding age group of voluntary evacuees (forced: 53.9%; voluntary: 17.6%; χ^2^ test, *p* < 0.01), whereas the rates in those aged ≥65 years were similar between forced and voluntary evacuees (forced: 57.8%; voluntary: 52.6%; χ^2^ test, *p* = 0.69) (Table 1 and Appendix A). This result might be explained by differences in the age group composition of forced and voluntary evacuees.

### 4.2. Specific Aspects of Current Strong Health Anxiety

The proportion of respondents with current strong health anxiety due to radiation exposure at the time of answering the questionnaire was 20.3%. Among evacuees who experienced current strong health anxiety at the time of answering the questionnaire, more than 90% were concerned about the delayed effects, the genetic effects, and broadcasting about nuclear issues. In previous studies, specific anxiety was associated with the effects of radiation on the development of thyroid cancer [13,14], on the workplace environment [15], on expectant mothers and children [16], on the estimated occurrence of acute radiation syndrome (an acute illness caused by irradiation of the entire body by a high dose of radiation in a short period of time) [17], and on the reluctance to eat foods grown in the evacuation area [18]. Moreover, among the Fukushima nuclear disaster evacuees, concerns about radiation risks were associated with psychological distress [19]. Although risk perception or anxiety regarding the delayed and genetic effects due to radiation exposure decreased from 2012 to 2015 (delayed effects: 48.1% in 2012 to 42.8% in 2015; genetic effects: 60.2% in 2012 to 37.6% in 2015) [18,19], these rates of risk perception and anxiety were still over 30% among all evacuees, even four years after the disaster. Therefore, despite the gradual decrease in the risk perception of radiation exposure, anxiety regarding the delayed and genetic effects due to exposure was associated with current strong health anxiety.

During the nuclear emergency in Fukushima, the traditional media were found to provide a broad context, including frequent comparisons with previous nuclear accidents; however, the experts’ technical vocabulary concerning radiation appeared incompletely translated for public understanding [20]. Therefore, our findings may show that, among evacuees who experienced current strong health anxiety at the time of answering the questionnaire, more than 90% had concerns about broadcasting regarding nuclear issues.

### 4.3. Association between Current Strong Health Anxiety and Utilization of Media Information: Considerations for Effective Disaster Communication

Among the responders, the proportion with current strong health anxiety due to radiation exposure at the time of answering the questionnaire was 20.3%. Current strong health anxiety at the time of answering the questionnaire was significantly lower among those who utilized public relations information from local government. The Public Relations Society of America (PRSA) stated that “Public relations is a strategic communication process that builds mutually beneficial relationships between organizations and their publics. It is thought that not providing information unilaterally but providing information promoting mutual-communication or useful information about lots of variety of consultation will lead to mutually beneficial relationships” [21]. In a previous report related to the Fukushima Daiichi Nuclear Power Plant accident, a unified approach was found no longer to be sufficient to address personal problems and anxiety as diverse information became available and people’s perceptions developed. This led to the need for one-to-one or small-group communication [22]. Another study reported that attending radiation information seminars or programs helped to reduce anxiety and psychological distress in a post-Fukushima disaster setting [12,23]. The public relations information from local government in evacuation areas included several articles regarding the health effects of radiation exposure, as well as the maximum annual exposure dose. Other posted articles included records of decontamination processing, discussion records of the health risk communication promotion committee in evacuation areas, and articles providing information such as general health consultations and dialogue among evacuees and experts, as well as information including education regarding stress reactions and coping [24]. Therefore, utilizing public relations information from local government may be associated with lower current health anxiety at the time of answering the questionnaire.

Users of Internet media tended to feel anxiety toward perceived radiation health risks, but not to a significant degree. Several studies have examined the correlation between risk perception and anxiety and media/information after the Fukushima nuclear accident. Murakami et al. revealed that dread risk perception was greater among people who trusted direct information from online researchers or others than those who did not, but was lower among people who trusted central governmental information than among those who did not [25]. Sugimoto et al. surveyed 1560 residents of Soma City in July 2011 and found that health anxiety was high among those who relied on word-of-mouth or rumors as a means to obtain information [12]. Baseless rumors and conspiracy theories spread very quickly on the Internet, which may also explain the high levels of anxiety among those who mainly used the Internet as their information source. Any Internet media usage in this study did not include solely Internet news, but also the use of personal websites such as social networking sites (SNS), and information from these sources likely include word-of-mouth or rumors. This suggests a potential association between evacuees with strong anxiety and the use of any type of Internet media.

### 4.4. Limitations and Strengths

This study has several limitations. First, due to its cross-sectional design, causality could not be established. Second, our primary outcome, i.e., current anxiety regarding perceived radiation health risks, was a subjective response This could certainly be the case with actual and perceived health risks, leading to important differences in anxiety levels, without validated measures and reliability statistics such as a test–retest correlation, which is a critical limitation. Therefore, it would hardly be applicable to different settings without reliability. Further studies are needed to confirm the validity and reliability regarding current anxiety regarding perceived radiation health risks. Additionally, our definition of “current strong health anxiety” may be included as a bias. Those who responded to “Somewhat” (47.1% in both the forced and the voluntary evacuees, see Appendix A) as current health anxiety at the time of answering the questionnaire due to radiation exposure were categorized as the “no or weak health anxiety” group, although this could be in either group. However, this categorization was comparable to that of the Fukushima Health Management Survey report [26], which focused on high and extreme anxiety of health effects due to radiation exposure. The third limitation relates to sampling from non-evacuation areas. We could not grasp detailed enough information about the number of voluntary evacuations in advance. As a result, many residents had not experienced evacuation voluntary, and thus more than 600 respondents were later excluded from the analysis. Fourth, because respondents tended to be relatively older, our study population included fewer Internet users, in particular SNS users. Finally, depending on the three types of evacuation areas according to the spatial radiation dose rates, there may have been different perceived health risks, leading to important differences in anxiety levels. However, due to anonymous sampling, it was impossible to obtain the detailed information on whether subjects were living in one of the three types of evacuation areas. This is because the first-stage sampling was selected by municipality, not by each of the three area types.

Despite these limitations, we were able to show a positive association between the utilization of public relations information from local government and lower health anxiety due to radiation exposure, even after adjusting for age, gender, education, and evacuation type. Although we examined the association between utilization of media and current health anxiety after the 2011 nuclear disaster, further studies in other settings, such as that of the novel coronavirus pandemic, are needed.

## 5. Conclusions

The 2011 nuclear disaster in Fukushima was not only a health disaster, but also an information disaster. Our findings highlight the importance of public relations information from local government in terms of it being associated with lower current health anxiety at the time of answering the questionnaire related to disaster situations, and this could potentially aid in preparing for future disasters.

## Figures and Tables

**Figure 1 ijerph-17-03921-f001:**
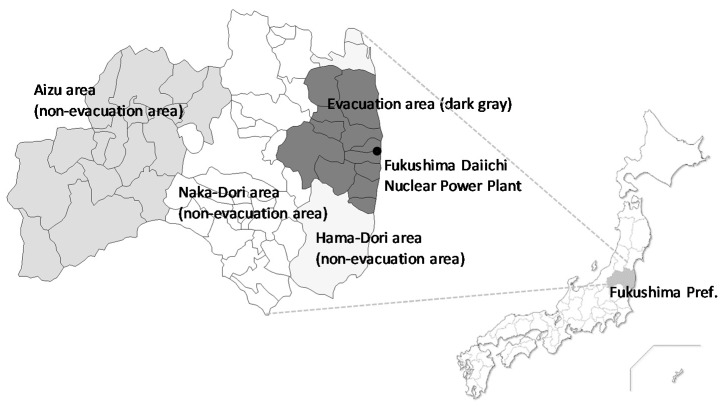
Evacuation and non-evacuation areas in Fukushima. Regions colored in dark gray correspond to the municipalities where evacuation orders were issued. Hama-Dori, Naka-Dori, and Aizu were the non-evacuation areas.

**Figure 2 ijerph-17-03921-f002:**
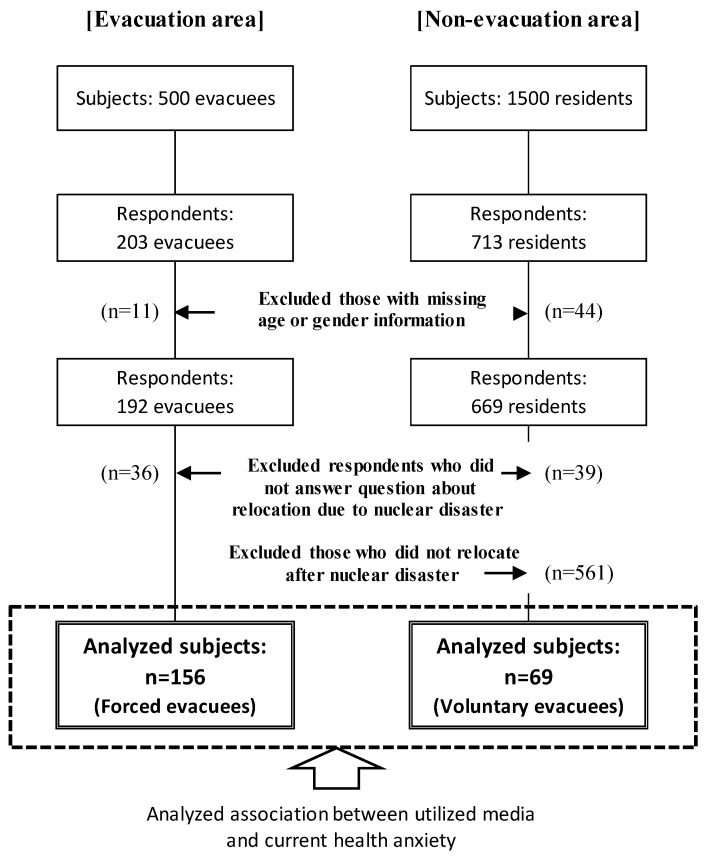
Sample selection in the evacuation and non-evacuation areas. The analyzed subjects included 156 forced evacuees and 69 voluntary evacuees.

**Table 1 ijerph-17-03921-t001:** Basic characteristics of the participants (forced/voluntary evacuees).

		Total		Forced Evacuees		Voluntary Evacuees		*p*-Value (χ^2^)
		(*n* = 225)		(*n* = 156)		(*n* = 69)		
		*n* (%)		*n* (%)		*n* (%)		
*Age (as of August 2016)*								
	<40 years	32	(14.2)	16	(10.3)	16	(23.2)	
	40–64 years	110	(48.9)	76	(48.7)	34	(49.3)	0.02 (χ^2^ = 7.99)
	≥65 years	83	(36.9)	64	(41.0)	19	(27.5)	
*Gender*								
	Male	89	(39.6)	61	(39.1)	28	(40.6)	0.83 (χ^2^ = 0.04)
	Female	136	(60.4)	95	(60.9)	41	(59.4)	
*Education*								
	Junior/senior high school	145	(65.6)	110	(71.9)	35	(51.5)	<0.01 (χ^2^ = 8.70)
	Vocational college, university, or graduate school	76	(34.4)	43	(28.1)	33	(48.5)	
*Occupational category*								
	Employed or owner	105	(47.5)	58	(37.9)	47	(69.1)	
	Suspended from job	7	(3.2)	7	(4.6)	0	(0.0)	<0.01 (χ^2^ = 19.5)
	Unemployed	109	(49.3)	88	(57.5)	21	(30.9)	
*Living area as of March 11, 2011*								
	Evacuation areas	156	(69.3)	156	(100.0)	0	(0.0)	<0.01
	Non-evacuation areas	69	(30.7)	0	(0.0)	69	(100.0)	(χ^2^ = 220.3)
	(Hama-Dori area)	50	(22.2)	-	-	50	(72.5)	
	(Naka-Dori area)	15	(6.7)	-	-	15	(21.7)	
	(Aizu area)	4	(1.8)	-	-	4	(5.8)	

**Table 2 ijerph-17-03921-t002:** Utilization of media relating to nuclear exposure among evacuees (forced/voluntary).

		Total		Forced Evacuees		Voluntary Evacuees		*p*-Value (χ^2^)
		(*n* = 225)		(*n* = 156)		(*n* = 69)		
		*n* (%)		*n* (%)		*n* (%)		
*Local media*								
	Local newspapers	140	(62.2)	102	(65.4)	38	(55.1)	0.14 (χ^2^ = 2.16)
	Local broadcasting	71	(31.6)	42	(26.9)	29	(42.0)	0.03 (χ^2^ = 5.05)
	Any local media	157	(69.8)	112	(71.8)	45	(65.2)	0.32 (χ^2^ = 9.81)
*National media*								
	National newspapers	29	(12.9)	23	(14.7)	6	(8.7)	0.21 (χ^2^ = 1.56)
	National broadcasting	39	(17.3)	26	(16.7)	13	(18.8)	0.69 (χ^2^ = 0.16)
	Any nationwide media	63	(28.0)	46	(29.5)	17	(24.6)	0.46 (χ^2^ = 0.56)
*Public broadcasting (NHK)*		102	(45.3)	76	(48.7)	26	(37.7)	0.13 (χ^2^ = 2.35)
*Internet media*								
	Internet news	33	(14.7)	16	(10.3)	17	(24.6)	0.01 (χ^2^ = 7.91)
	Other information on Internet	21	(9.3)	8	(5.1)	13	(18.8)	<0.01 (χ^2^ = 10.6)
	Social networking sites (SNS)	12	(5.3)	5	(3.2)	7	(10.1)	0.03 (χ^2^ = 4.56)
	Any Internet media	54	(24.0)	24	(15.4)	30	(43.5)	<0.01 (χ^2^ = 20.7)
*Public relations from local government*		99	(44.0)	79	(50.6)	20	(29.0)	<0.01 (χ^2^ = 9.11)

**Table 3 ijerph-17-03921-t003:** Characteristics of current anxiety (forced/voluntary evacuees).

	Total					Forced Evacuees					Voluntary Evacuees				
	Current Strong Anxiety about Health due to Nuclear Disaster					Current Strong Anxiety about Health due to Nuclear Disaster					Current Strong Anxiety about Health due to Nuclear Disaster				
	(+)		(−)		*p*-Value (χ^2^)	(+)		(−)		*p*-Value (χ^2^)	(+)		(−)		*p*-Value (χ^2^)
	(*n* = 43)		(*n* = 178)			(*n* = 29)		(*n* = 124)			(*n* = 14)		(*n* = 54)		
	*n* (%)		*n* (%)			*n* (%)		*n* (%)			*n* (%)		*n* (%)		
***Anxiety about delayed effects***	39	(92.9)	81	(45.5)	<0.01 (χ^2^ = 30.7)	25	(89.3)	54	(43.5)	<0.01 (χ^2^ = 19.1)	14	(100.0)	27	(50.0)	<0.01 (χ^2^ = 11.6)
***Anxiety about unhealthy status***	26	(63.4)	34	(19.2)	<0.01 (χ^2^ = 32.6)	17	(60.7)	26	(21.0)	<0.01 (χ^2^ = 17.8)	9	(69.2)	8	(15.1)	<0.01 (χ^2^ = 16.0)
***Anxiety about genetic effects***	39	(92.9)	84	(47.2)	<0.01 (χ^2^ = 28.7)	26	(92.9)	57	(46.0)	<0.01 (χ^2^ = 20.3)	13	(92.9)	27	(50.0)	<0.01 (χ^2^ = 8.43)
***Anxiety from broadcasting about a nuclear event***	41	(95.3)	129	(73.3)	<0.01 (χ^2^ = 9.68)	27	(93.1)	85	(69.7)	0.01 (χ^2^=6.72)	14	(100.0)	44	(81.5)	0.08 (χ^2^=3.04)

**Table 4 ijerph-17-03921-t004:** Association between current strong anxiety and media utilization (forced/voluntary evacuees).

	Total					Forced Evacuees					Voluntary Evacuees				
	Current Strong Anxiety about Health due to Nuclear Disaster					Current Strong Anxiety about Health due to Nuclear Disaster					Current Strong Anxiety about Health due to Nuclear Disaster				
	(+)		(−)		*p*-Value (χ^2^)	(+)		(−)		*p*-Value (χ^2^)	(+)		(−)		*p*-Value (χ^2^)
	(*n* = 43)		(*n* = 180)			(*n* = 29)		(*n* = 126)			(*n* = 14)		(*n* = 54)		
	*n* (%)		*n* (%)			*n* (%)		*n* (%)			*n* (%)		*n* (%)		
*Any local media*	31	(72.1)	124	(68.9)	0.68 (χ^2^ = 0.17)	22	(75.9)	89	(70.6)	0.57 (χ^2^ = 0.32)	9	(64.3)	35	(64.8)	0.97 (χ^2^ = 0.01)
*Any national media*	14	(32.6)	49	(27.2)	0.49 (χ^2^ = 0.49)	11	(37.9)	35	(27.8)	0.28 (χ^2^ = 1.16)	3	(21.4)	14	(25.9)	0.73 (χ^2^ = 0.12)
*Public broadcasting (NHK)*	15	(34.9)	85	(47.2)	0.14 (χ^2^ = 2.13)	11	(37.9)	64	(50.8)	0.21 (χ^2^ = 1.56)	4	(28.6)	21	(38.9)	0.48 (χ^2^ = 0.51)
*Any Internet media*	13	(30.2)	40	(22.2)	0.27 (χ^2^ = 1.23)	4	(13.8)	20	(15.9)	0.78 (χ^2^ = 0.08)	9	(64.3)	20	(37.0)	0.07 (χ^2^ = 3.38)
*Public relations from local government*	12	(27.9)	87	(48.3)	0.02 (χ^2^ = 5.87)	9	(31.0)	70	(55.6)	0.17 (χ^2^ = 5.67)	3	(21.4)	17	(31.5)	0.46 (χ^2^ = 0.54)

**Table 5 ijerph-17-03921-t005:** Multivariate logistic regression analysis with utilized media and current strong anxiety (forced/voluntary evacuees).

		Model 1			Model 2					
		Total Current Strong Anxiety (+/−)			Forced Evacuees Current Strong Anxiety (+/−)			Voluntary Evacuees Current Strong Anxiety (+/−)		
		(*n* = 219)			(*n* = 152)			(*n* = 67)		
		OR (95% CI)		*p*-Value	OR (95% CI)		*p*-Value	OR (95% CI)		*p*-Value
*Any local media*	Yes	1.11	(0.90–1.36)	0.35	1.11	(0.86–1.44)	0.42	1.07	(0.72–1.57)	0.75
	No (Ref.)	1.00			1.00			1.00		
*Any national media*	Yes	1.01	(0.82–1.25)	0.90	1.04	(0.81–1.34)	0.74	0.89	(0.57–1.38)	0.60
	No (Ref.)	1.00			1.00			1.00		
*Public broadcasting (NHK)*	Yes	0.85	(0.69–1.04)	0.11	0.83	(0.65–1.05)	0.12	0.88	(0.59–1.31)	0.53
	No (Ref.)	1.00			1.00			1.00		
*Any Internet media*	Yes	1.11	(0.88–1.43)	0.36	0.96	(0.68–1.35)	0.81	1.56	(0.99–2.43)	0.05
	No (Ref.)	1.00			1.00			1.00		
*Public relations from local government*	Yes	0.76	(0.61–0.94)	0.01	0.72	(0.56–0.93)	0.01	0.89	(0.59–1.34)	0.57
	No (Ref.)	1.00			1.00			1.00		

Model 1: Adjusted for gender, age, education, and evacuation type. Model 2: Adjusted for gender, age, and education OR, odds ratio; CI, confidence interval.

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
