# Peer review of "The Association between Utilization of Media Information and Current Health Anxiety Among the Fukushima Daiichi Nuclear Disaster Evacuees"

_ijerph, 2020, doi:10.3390/ijerph17113921_

Round 1
Reviewer 1 Report
I’ve read with attention the paper of Orui et al. that is potentially of interest. The background and aim of the study have been clearly defined. The methodology applied is overall correct, the results are reliable and adequately discussed. I’ve only some minor comments: - Among study limitation, the authors should consider that their results are directly applicable only to the Japanese population and in particular the one of the Fukushima Region. Even the results are suggestive, but they would be hardly applicable to different settings, without a confirmation
- Probably, the authors should infer about the relevance of an adequate comunication by the institutions also in other context like this (I mean, for instance, in epidemic conditions, like the last one we are living now).
Author Response
Thank you for your comments. I would like you to see the attached file.
Masatsugu Orui

Reviewer 2 Report
Main comment: Although the title of this manuscript is “The Association ….”, the authors tend to make some “causal” comments. It should be carefully not to interpret “the utilization of media information” as a cause or result of the “health anxiety”. Carefully revising the paper to remove such discussions is suggested.
Other comments:
- (Section 4.1. Utilization of media information regarding the effects of radiation on health) Comment: I fail to see any discussion regarding the effect of radiation on health in this subsection. The authors compared the distribution of media utilization by age group and evacuation type. Does the “evacuation type” represent “the effects of radiation on health” in the study? Furthermore, the findings in this subsection are a bit obvious. For examples, “forced evacuees” were arranged by government ==> they need to follow info from local government, and they will maintain original community structure after relocation; “voluntary evacuees” were arranged by themselves ==> they needn’t follow info from local government closely, and they will where to relocate by themselves.
- (Lines 253-255) “although not significant, Internet media users tended to have strong health anxiety regarding radiation exposure”
Comment: It is inappropriate to present this as a causal relation: heavy Internet media users ==> strong health anxiety. In fact, it is more likely to be the other way around: strong health anxiety ==> heavy Internet media users. Those with stronger health anxiety tried to get as more info as possible, and consequently they use more internet media than those with less anxiety as a result. This also explains why those only relied on the info from local governments showed the least anxiety.
- Comment: The term “any internet media” may be a bit confusing in this paper. The lines between traditional and internet media have been blurred as all traditional media content can be accessed online.
- (Line 325-326) Comment: Remove the redundant “new-line”.
Author Response

(The authors gave the same response as above.)

Reviewer 3 Report
I congratulate the authors on a sound scientific study of high interest to the general readership of IJERPH. I suggest a minor review and have provided detailed comments in order to help the authors improve the structure, logic and flow of their argument and discussion.
It was an honor to read and review a manuscript on such an important and topical subject.
Abstract:
- Please include the total number of study participants and sub-group population sizes in your abstract.
- Please do not report statistically non-significant results in your abstract.
- Please redraft the first line of the abstract and delete "perceived health risks". This nuclear disaster did have major health impacts on the population and the environment.
- Lines 57-58 from the introduction: This is a brilliant sentence that puts your entire study into context. I strongly urge the authors to include this as one of the first sentences in the abstract and the introduction.
- Line 76 from the introduction: The authors refer to "current strong anxiety" but what do they mean by current? What year was the cross-sectional study conducted? And far apart is the timepoint of the study from the nuclear disaster. This is a crucial point and should be included in the abstract as well.
- Lines 89-91 introduction. If word count permits, please include the sampling strategy in the abstract.
Introduction:
Line 45: Here you state that residents were forced to leave their home but in your abstract and your methods you differentiate between evacuees who left voluntarily and those who were forced. Clarify this in the first part of your introduction please. For example, in lines 132-135 you list evacuation types and provide descriptions. Can some of that information be included here?
Line 48: Perhaps better worded if you say that one of the public health risks was increased anxiety and mental health issues due to perceived risk...this is very different than saying perceived risk as a public health issue.
Line 57: Who or What is Yamashita - best not to assume that your reader is informed. Please be more specific.
Lines 57-58: This is a brilliant sentence that puts your entire study into context. I strongly urge the authors to include this as one of the first sentences in the abstract, introduction and revisit this sentence in the discussion, conclusion. The current placement of this sentence comes too late in the argument here and is better placed further up.
Line 76: The authors refer to "current strong anxiety" but what do they mean by current? What year was the cross-sectional study conducted? And far apart is the timepoint of the study from the nuclear disaster. This is a crucial point and should be included in the abstract as well.
Materials and Methods:
Line 86: How were the study participants selected? Randomly? Convenience? Were surveys sent to a larger group of evacuees but only 500 surveys were returned and used in the study? Please be specific. I see now, that the sampling strategy is included on lines 89-91. Please move it here.
Line 86: The authors refer to "the evacuation area" but in the previous sentence listed 3 different evacuation areas. Is the current study population a sample from all 3 areas? If so, what was the distribution of the study population across the 3 areas. This could certainly actual and perceived health risk leading to important differences in anxiety levels.
Line 87: This is the first mention of the areas by name and I am unsure how these areas relate to those mentioned in line 86. Please clarify.
Lines 89-91. The sampling strategy here needs to be the first line of this section. Please move up, and also include in the abstract word count permitting.
Line 102: By current anxiety, do the authors refer to anxiety levels in 2016? If so, please state this throughout the manuscript.
Line 118: The authors state, "regarding radiation anxiety" but could be more specific and state "health anxiety due to radiation exposure" or something of the sort. Do not be worried about repeating words or phrases. The reader needs consistency in phrasing, definitions and meanings.
Lines 132-135: Here you list evacuation types and provide descriptions. Please move this section or a modified version of this section further up in your paper.
Results:
Lines 146-147. As a secondary analysis for a future paper, it would be interesting to know the results from the respondents who were not included in this study (After excluding 55 respondents who failed to provide information regarding sex or age, as well as 636 respondents). this is valuable information with a high participation rate (higher than the current population in the current study) and could definitely make for an interesting second paper. In fact, the authors could even compare the two populations (current study population compared to 55+636 respondents that were excluded from the current study).
Line 157: Table 1: Living areas: The study population in Hama-Dori, Naka-Dori, Aizu areas are too small to be seperated and should be combined for greater statistical power to detect meaningful differences.
Line 172: What is the proportion of all study participants who expressed current anxiety? Please add that number here. If the vast majority of your participants had current anxiety, then this is a major finding. For example, if 92% of 98% were concerned about delayed or genetic effects, this is much more significant if that same 92% only represented 50% of your overall study population.
Discussion
Line 201: I am not convinced that the first major finding listed in this section should be considered a major finding at all; "significant difference in utilization of Internet media and public relations information from 201 local governments between forced and voluntary evacuees".
Line 233: Please include the total proportion of all study participants who expressed current anxiety (see my comment in results, line 172).
Line 244: Please include some numbers here, "decreased from 2012 to 2015"
Lines 248-249. Please replace "As results" with "Therefore" or "As such" or a similar phrase.
Line 293: Please insert "perceived" before "radiation health risk" and please check this consistency throughout the manuscript.
Line 302: Recall bias of what? Recall bias on media usage at time of the disaster? Please be specific.
Lines 303-304: I do not think your response issue was a problem. I think your sampling was an issue considering you had more than 600 responses from indviduals who were later cut from the analysis and 55 who did not report age or sex.Please adjust this limitation accordingly.
Line 309: Here you only mention one major finding which brings me back to my earlier concern from Line 201. Please see my comment above.
Conclusion
Lines 322-324. Please revise this conculsion statement. In fact, your conclusion is that providing appropriate health communication in a disaster could mitigate health anxiety among community members. But please use your own words and also be sure this is reflected in your abstract.
Author Response

(The authors gave the same response as above.)
